# Ecological assessment of water quality in freshwater wetlands based on the effect of environmental heterogeneity on phytoplankton communities in Northeast China

Hongkuan Hui[1]*, Xiao Liu[1], Yinxin Wei[2], Dan Su[3], Haitao Zhou[3], Zirui Peng[1]

1 School of Geography and Tourism, Qilu Normal University, Jinan, China, 2 Institute of Hydrobiology, Chinese Academy of Sciences, Wuhan, China, 3 Heilongjiang Province Key Laboratory of Geographical Environment Monitoring and Spatial Information Service in Cold Regions, Harbin Normal University, Harbin, China

* 20157377@qlnu.edu.cn

**Data Availability Statement:** All relevant data are within the manuscript and its Supporting Information files.

## Abstract

Phytoplankton community characterized by strong vitality response to environmental change in freshwater ecosystems. This study aims to evaluate the effectiveness of using phytoplankton diversity as a water quality indicator in wetlands, and find out the main environmental variables affecting the distribution of phytoplankton. From 2020 to 2021, we examined phytoplankton assemblages and water environmental variables in spring, summer, and autumn at eight sampling sites from Hulanhe Wetland, Northeast (NE) China. The results showed that Bacillariophyta was the dominant species. Phytoplankton composition and abundance differed among sampling sites in each season; the abundance in summer ($613.71 \times 10^4$ ind. $L^{-1}$) was higher than that in autumn and spring. The water quality assessment of the trophic state index (*TSI*) based on the four physicochemical indicators was compared with phytoplankton diversity indices, which indicated that the phytoplankton community was stable, and these two indices were significantly lower in summer than in spring and autumn. According to redundancy analysis (RDA), total phosphorus (TP) and nitrogen (TN) were the main environmental variables affecting the distribution of phytoplankton. Temperature and dissolved oxygen (DO) changes also played a role, and their impact on the community was discussed. This work can provide relevant scientific references on the usefulness of phytoplankton diversity structure in assessing water quality in cold regions, in which the succession can be significantly affected by nutrients and temperatures.

## Introduction

Phytoplankton is a primary producer of aquatic food webs in wetlands, the main starting point of aquatic ecosystems, and a high-quality feed for zooplankton and other aquatic organisms [1]. Due to the characteristics of numerous species, wide distribution, small size, and short life

**Funding:** This work was supported by the Natural Science Foundation of Shandong Province (ZR2023QC238, ZR2023QC253), the Project of Shandong Province Higher Educational Science and Technology Program (No. J18KA199). There was no additional external funding received for this study.

**Competing interests:** The authors have declared that no competing interests exist.

cycle of phytoplankton, the community structure formed is easily influenced by environmental and biological factors (feeding, interspecific competition, etc.). For some species that have advantages in the ecological community structure, their species and cell numbers are usually important indicators of changes in the ecological environment of water bodies [2, 3].

Because of their sensitivity to environmental conditions, phytoplankton provide information on the quality of habitats and can reflect changes in different environmental gradients. It is important to evaluate the physical, chemical, and biological processes of aquatic ecosystems by investigating the phytoplankton populations [4–6]. Therefore, phytoplankton have been widely used as an indicator of the nutritional status of aquatic environments and have increasingly been used to assess water quality, in particular, to monitor the eutrophication of lakes [7–9]. However, in recent years some works have pointed out that using only one diversity index to explain the diversity of phytoplankton communities can easily lead to significant deviations, and different opinions have been proposed on the bio-geographical factors of phytoplankton diversity about water quality [10–12].

Phytoplankton diversity is influenced by the aquatic environmental conditions [13, 14]. Several studies have shown that environmental factors such as water temperature, pH, water chemistry, and nutrient loading can affect the species composition in certain phytoplankton communities [7, 15, 16]. In addition, the response of phytoplankton to the changes in water quality, such as increased nutrient loading can be rapid. The trophic state index (*TSI*) is currently the most commonly used method for evaluating the nutritional status of surface waters, which is a continuous grading method based on the sensitivity of a single benchmark and the comprehensiveness of multiple auxiliary factors. The Chl. *a* is the final indication due to its direct manifestation of eutrophication-related risk, and the physiochemical indicators (total nitrogen, total phosphorus, and permanganate index) are indirect indications [17–19]. The differences in hydrological morphological backgrounds are likely to correspond to complex and variable response relationships between Chl. *a* and physicochemical indicators (TN, TP, and COD). Therefore, whether biological indicators or *TSI* (Σ) indicators are used alone, they cannot accurately evaluate water ecosystems [20, 21].

Studying the level of species diversity and the preferred environmental conditions for phytoplankton in wetlands can also provide an opportunity to find out the relationship between phytoplankton and environmental factors and assess the ecological health of this particular ecosystem [22–24]. It has been established that it is essential to investigate the relationship between the distribution of phytoplankton and aquatic environment in lakes, rivers, and wetlands [25, 26].

The specific objectives of this study were to investigate the effectiveness of phytoplankton diversity as a water quality indicator in wetland ecosystems. To verify the hypothesis that environmental factors may have a greater impact on phytoplankton diversity compared to water quality parameters, we analyzed the spatial and seasonal distribution of phytoplankton assemblages and assessed the water quality by analyzing the physiochemical parameters, *TSI* (Σ), and phytoplankton diversity in the Hulanhe Wetland, Northeast China. We then identified the main environmental factors which affect the phytoplankton communities. Our study will guide for using phytoplankton as an indicator to understand environmental change and basic ecological information for future research in similar wetlands.

## Materials and methods

### Study sites and period

The Hulanhe Wetland is located in the south of the Hulan District, Harbin City, Heilongjiang Province, on the north bank of the Songhua River. The study area is in the center of the

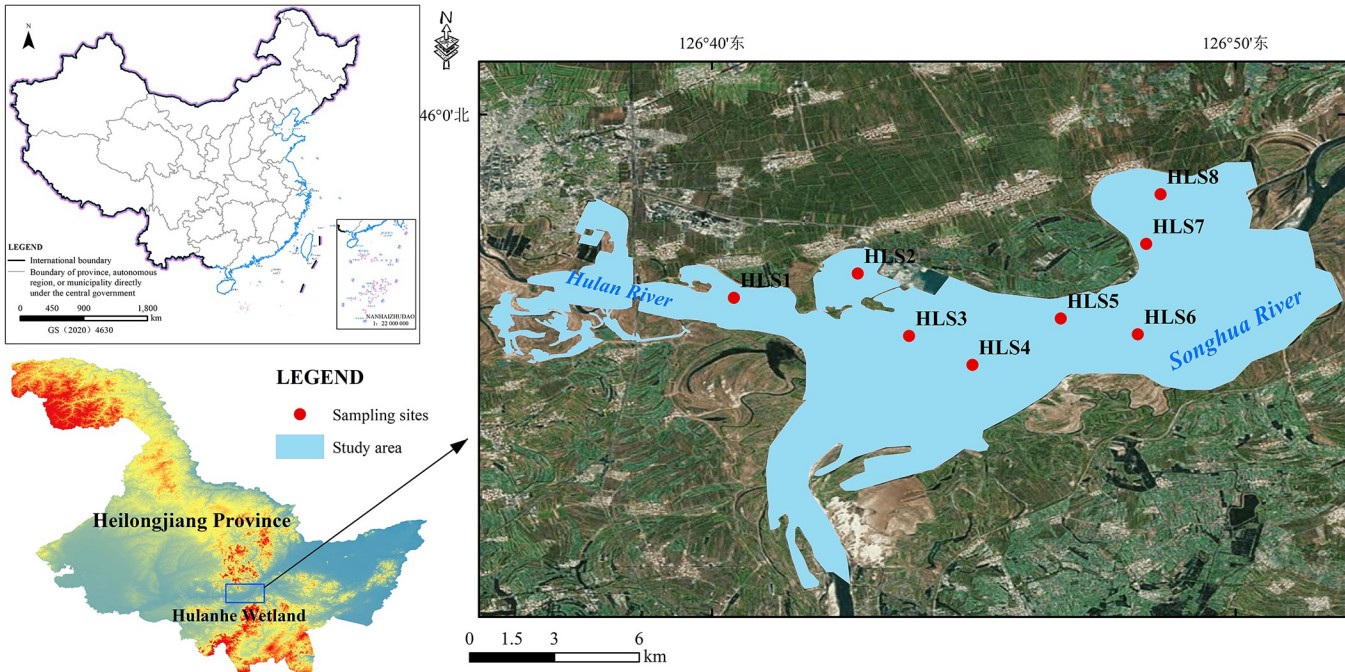

**Fig 1. Location of the study area and sites at Hulanhe Wetland.**

wetlands between 126°38′- 127°14′ E and 45°5′- 45°51′ N, covering an area of 192.62 km². The wetlands belong to the continental climate of the north temperate zone, with four distinct seasons, long winters and short summers, and the rapid rising and falling of temperatures in the spring and autumn.

Using the Global Positioning System (GPS), eight sampling sites (HLS1–HLS8) were selected across the Hulanhe Wetland from May to October 2020 and 2021, respectively (Fig 1). According to the flow direction of the Hulan River after entering the Songhua River, the wetland Hulanhe Wetland is divided into four ecological water bodies. The upstream intake area (UIA) includes site HLS1 and HLS2, located at the first of hydrological monitoring station, representing the semi-agricultural area. The wetland conservation area (WCA) includes site HLS3 and HLS4, they are in the open water and not subject to the impact of sewage discharge, which is far away from villages and has much faster water flow (the average flow velocity more than 0.10 m s⁻¹). Site HLS5 and HLS6 belong to the environmental monitoring area (EMA), near the village but no sewage enters the water here. Site HLS5 is located at the second of the hydrological monitoring stations. The farmland and residential area (FRA, sites HLS7 and HLS8) are located near the outfall of a village, which means these waters are disturbed by human activities. This is an enclosed area of water (a pond) and a large area of farmland surrounds this area.

## Physical and chemical analysis

A total of 96 water samples (48 for 2020 and 48 for 2021) were collected from Hulanhe Wetland. Conductivity, pH, and water temperature at each site were measured in the field using a hand-held Manta 2 (Eureka, USA). Approximately 500 mL of water was collected and placed in polyethylene bottles for chemical analysis in each sampling site monthly. All water chemistry measurements followed standard methods and were analyzed within 24 h of collecting the

samples [27]. Dissolved oxygen (DO) determinations were carried out using the iodometric method. Total phosphorus (TP) and nitrogen (TN) determinations were carried out using a UV-Vis spectrophotometer (UV-2500, Shimadzu Corporation, Japan). For total phosphorus (TP), we used the molybdenum blue method and measured the absorbance at wavelength 700 nm. Total nitrogen (TN) was analyzed with potassium peroxide sulphate and absorbance was measured at wavelengths 220 and 275 nm. Total organic carbon (TOC) was determined using the combustion oxidation-non-dispersive infrared absorption method.

The formula for calculating the comprehensive Trophic State Index (*TSI*) based on water quality physicochemical indicators (TN, TP, and COD) is as follows:

$$TSI(TN) = 10 \times (5.453 + 1.694 \ln TN)$$

$$TSI(TP) = 10 \times (9.436 + 1.624 \ln TP)$$

$$TSI(COD) = 10 \times (0.109 + 2.661 \ln COD)$$

$$TSI(\sum) = [(TN) + (TP) + (COD) + (BOD)]/4$$

Based on the *TSI* (Σ) values, the water quality was classified into five states: oligotrophic (0–30), mesotrophic (30–50), light eutrophic (50–60), middle eutrophic (60–70), and hypertrophic (70–100).

## Phytoplankton sampling and analysis

A total of 96 phytoplankton samples (48 for 2020 and 48 for 2021) were collected from Hulanhe Wetland. The samples were collected from the upper layer of the water up with a plankton net (mesh size 50 μm) and were preserved using Lugol's solution. For quantitative analyses of phytoplankton communities, 1000 mL mixed water was collected at each site and precipitated for 24–36 h. Then, the final volume was set to 30 mL of concentrated sediments. A 0.1 mL subsample of this final concentrate was placed in a perspex counting chamber (Uwitec, Austria) and enumerated under a light microscope at a magnification of 400× (Olympus BH×2, Japan). The figure with mean values as its biological abundance and 500 valves were counted from each sample. Identification of taxa followed the volumes by Hu and Lange-Bertalot [28, 29].

The Shannon-Weaver index (*H′*) and Pielou evenness index were used to explain species diversity for phytoplankton calculation [30]. The species diversity calculation was performed using the following formulas:

$$\text{Shannon−Weaver index}(H') : \qquad H' = -\sum \left(\frac{ni}{N}\right) * \log_2 \left(\frac{ni}{N}\right)$$

$$\text{Pielou evenness index}(J) : \quad J = \frac{H'}{\ln S}$$

where *ni* represents an individual amount of the phytoplankton species, *N* is the total individual number of phytoplankton, and *S* is the total number of species. For the Shannon-Weaver index (*H′*), 0–1 indicates polysaprobic (heavily polluted water), 1–3 indicates mesosaprobic (moderately polluted water), and more than 3 indicates oligosaprobic (clean water). For the evenness index (*J*), the values of the index vary from 0 to 0.8 and indicate water quality (0–0.3, polysaprobic; 0.3–0.5, mesosaprobic; 0.5–0.8, oligosaprobic) [31].

## Statistical analysis

Analysis of variance (ANOVA, SPSS22.0) was used to test differences between sites with respect to water environmental variables, species richness, and diversity. Spatial distribution mapping of phytoplankton abundance using ordinary Kriging was performed with the ArcGIS 10.8, commercial geographic information system software developed by ESRI Co, Redlands, USA. To investigate the relationships between phytoplankton and water environmental variables, a redundancy analysis (RDA) was performed [32]. In this paper, the calculation process and drawing were conducted using Canoco 5.0 for Windows. All environmental factors were normalized using the formula log (1+x) except pH [33, 34]. Then, environmental factors were selected by the forward selection with Monte Carlo permutation tests (N = 999) for confidence interval analysis ($P < 0.05$).

## Results

### Environmental variables

The ranges of physical-chemical parameters at the eight sampling sites for a twelve-month baseline survey were summarized in Table 1. During two years, seasonal water temperature changed significantly and ranged from 4.3 to 27˚C; temperature was higher in 2021 than in 2020, but temperature showed minor differences at all sampling sites during our study period. DO ranged from 3.6 to 16.8 mg L$^{-1}$ (8.88 ± 0.52 mg L$^{-1}$), the concentrations showed significant differences in different seasons ($P<0.05$), spring and autumn were higher than summer and higher than 3 mg L$^{-1}$ at all eight sites, with the minimum average value at HLS5 (7.71 ± 1.40 mg L$^{-1}$) and the maximum at HLS3 (9.40 ± 1.51 mg L$^{-1}$). High concentrations of both COD and BOD occurred at HLS1 and HLS5, with the maximum in May and the minimum in October. The concentrations of TP and TN showed the same trends at all sampling sites in different seasons between 2020 and 2021: higher values at FRA compared to others. The TOC average concentration value of 4.4 mg L$^{-1}$ in 2021, was slightly lower than in 2020, with an average value of 4.5 mg L$^{-1}$, whereas the concentrations in HLS1 (4.51 ± 0.47 mg L$^{-1}$), HLS7 (4.52 ± 0.48 mg L$^{-1}$), and HLS8 (4.53 ± 0.45 mg L$^{-1}$) were higher than other sites ($P < 0.05$). The concentration of conductivity ranged from 175 to 509 μs cm$^{-1}$, and the conductivity at HLS8 was obviously higher than that at other sites in the two years (490.75 ± 17.56 mg L$^{-1}$) (Fig 2).

Based on the *TSI* (Σ) values, the water quality of the Hulanhe Wetland is in mesotrophic state (Table 2, Fig 3). The *TSI* (Σ) values ranged from 38.24 to 49.26 in spring with a mean of 45.37, from 36.43 to 47.51 in summer with a mean of 42.84, and from 38.66 to 49.88 with a

**Table 1. Environmental variables at eight sampling sites in Hulanhe Wetland.**

|  | Spring | Summer | Autumn | Independent-samples *t*-test | | |
|---|---|---|---|---|---|---|
|  | Mean ± SD | Mean ± SD | Mean ± SD | Spring × Summer | Spring × Autumn | Summer × Autumn |
| pH | 6.57 ± 0.67 | 6.65 ± 0.66 | 6.68 ± 0.45 | *P > 0.05* | *P > 0.05* | *P > 0.05* |
| WT (˚C) | 16.75 ± 7.02 | 20.13 ± 7.05 | 14.33 ± 7.73 | *P < 0.05* | *P < 0.05* | *P < 0.05* |
| DO (mg L$^{-1}$) | 9.07 ± 5.25 | 8.58 ± 5.10 | 9.00 ± 5.19 | *P < 0.05* | *P < 0.05* | *P > 0.05* |
| BOD (mg L$^{-1}$) | 1.14 ± 0.47 | 2.22 ± 0.48 | 1.15 ± 0.43 | *P < 0.05* | *P > 0.05* | *P < 0.05* |
| COD (mg L$^{-1}$) | 10.06 ± 2.11 | 19.37 ± 1.96 | 14.06 ± 1.88 | *P < 0.05* | *P < 0.05* | *P > 0.05* |
| TN (mg L$^{-1}$) | 1.41 ± 0.24 | 1.65 ± 0.26 | 3.43 ± 1.29 | *P > 0.05* | *P < 0.05* | *P < 0.05* |
| TP (mg L$^{-1}$) | 0.09 ± 0.04 | 0.09 ± 0.05 | 0.09 ± 0.05 | *P > 0.05* | *P > 0.05* | *P > 0.05* |
| TOC (mg L$^{-1}$) | 4.45 ± 0.15 | 4.45 ± 0.16 | 4.44 ± 0.15 | *P > 0.05* | *P > 0.05* | *P > 0.05* |
| SpCond (μs cm$^{-1}$) | 189.87±32.01 | 187.56 ± 34.48 | 306.90 ± 56.24 | *P > 0.05* | *P < 0.05* | *P < 0.05* |

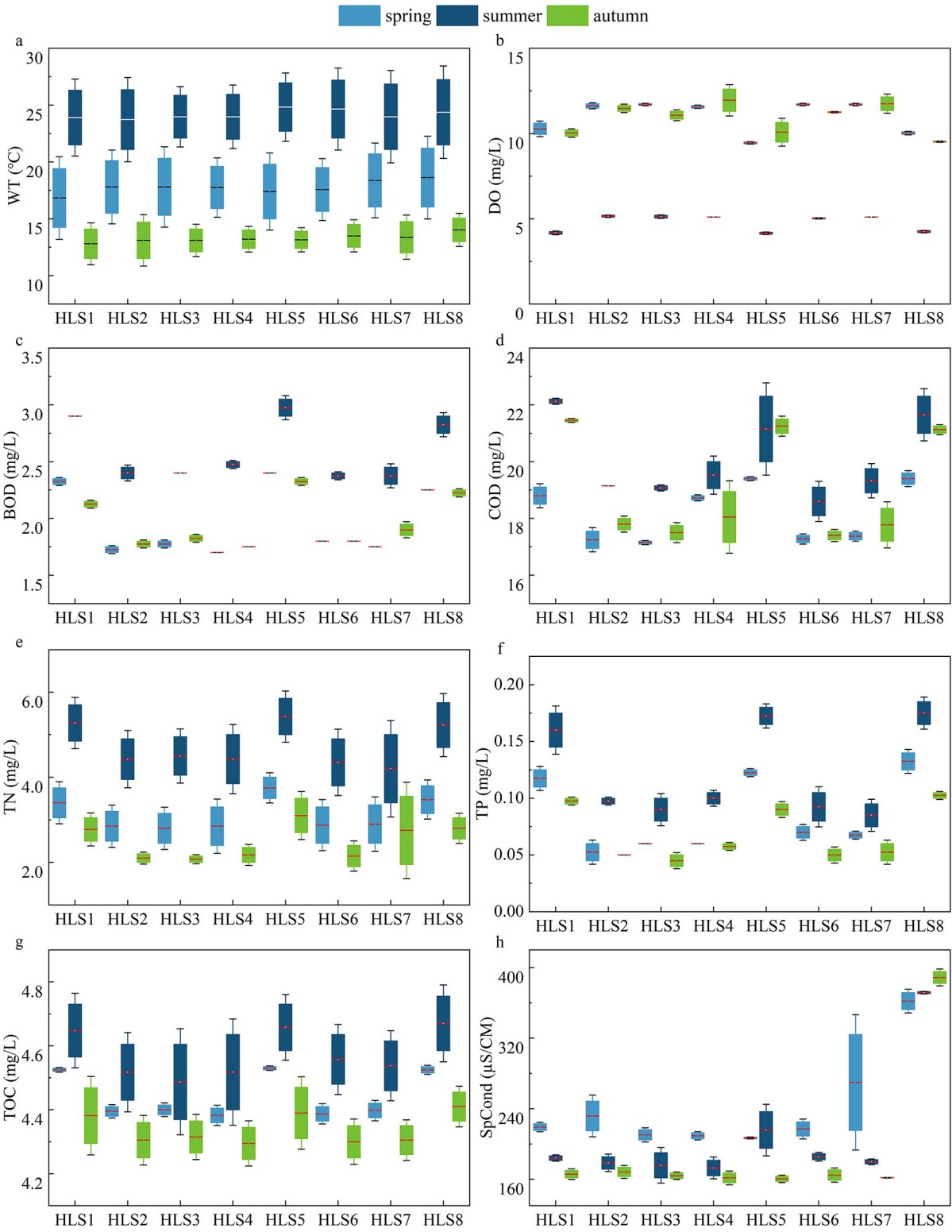

**Fig 2. Temporal and spatial variations of the main environmental variables in Hulanhe Wetland.**

**Table 2. The values of *TSI*(BOD), *TSI*(COD), *TSI*(TN), *TSI*(TP), and *TSI*(Σ) of the Hulanhe Wetland.**

| Area | *TSI* (BOD) | *TSI* (COD) | *TSI* (TN) | *TSI* (TP) | *TSI* (Σ) | P |
|------|-------------|-------------|------------|------------|-----------|---|
| UIA | 41.18 ± 5.15 | 49.88 ± 2.90 | 44.43 ± 3.12 | 34.44 + 2.43 | 43.12 ± 1.26 | < 0.05 |
| WCA | 38.79 ± 5.34 | 78.23 + 2.21 | 42.91 + 2.35 | 38.30 + 3.85 | 40.23 + 2.17 | < 0.05 |
| EMA | 41.93 + 4.99 | 79.53 + 1.23 | 45.12 + 1.63 | 35.04 + 1.16 | 45.60 + 1.77 | < 0.05 |
| FRA | 44.43 + 4.51 | 80.23 + 2.56 | 44.97 + 3.45 | 35.49 + 2.18 | 46.99 + 1.79 | < 0.05 |

mean of 46.14 in autumn, and the "contribution" of *TSI* (COD) to water eutrophication is significant (P<0.05) (S1 Table), with an average of 59.5 during the survey period. *TSI* (Σ) values of sampling sites HLS1, HLS7, and HLS8 are relatively high, with scores of 50.42, 49.94, and 50.17, respectively. The water quality near sampling sites HLS3 and HLS4 in WCA is good, with *TSI* (Σ) value of 40.23.

## Structure of the phytoplankton community

During the study period, a total of 216 species representing 50 genera were found in the study area. The composition was formed by Bacillariophyta (98 species), Chlorophyta (57 species),

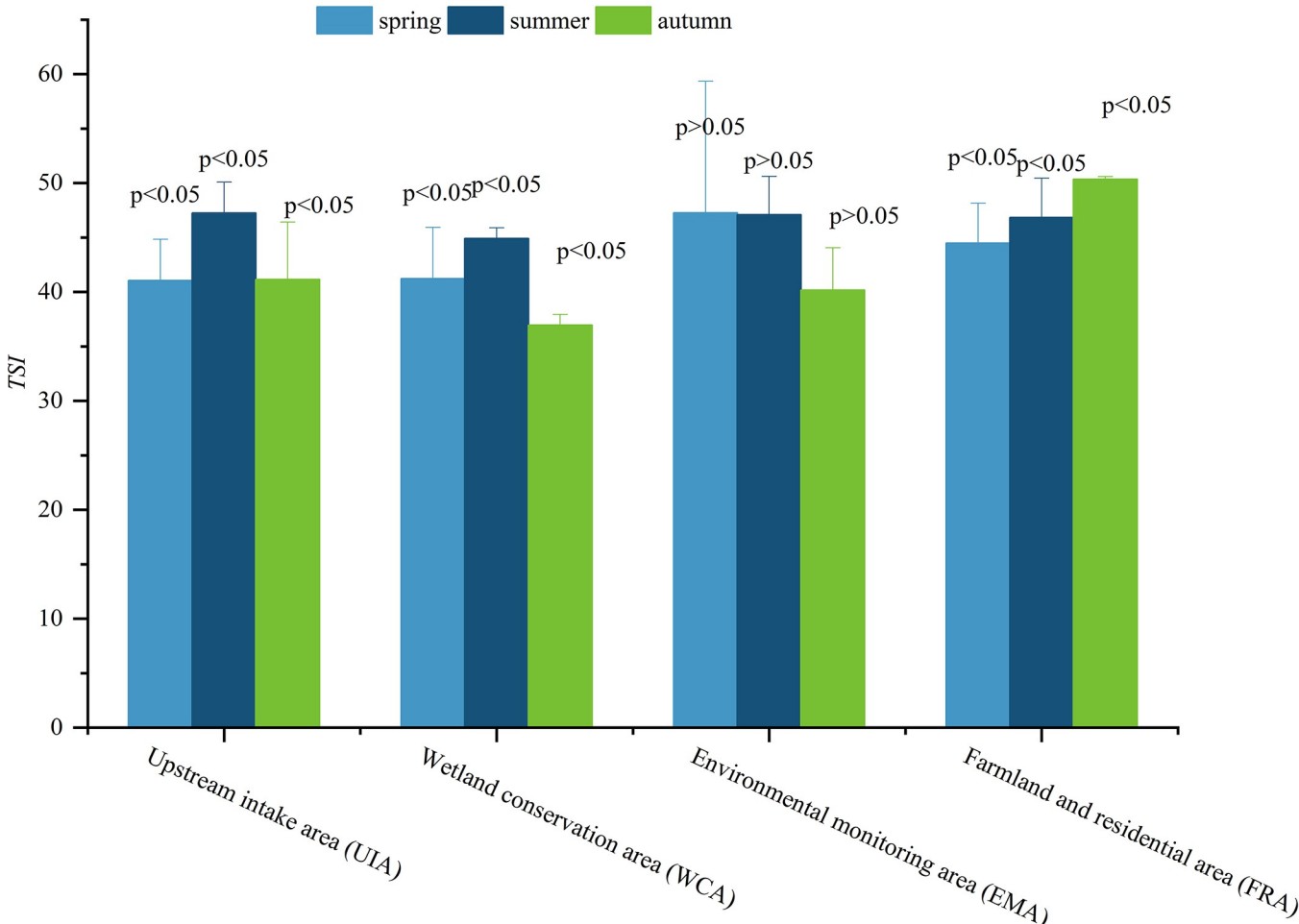

**Fig 3. The comprehensive trophic state index at four study area in Hulanhe Wetland.**

Table 3. Dominance factor values of the seasonal phytoplankton dominant species in Hulanhe Wetland.

| Code | Species | Spring | | Summer | | Autumn | |
|---|---|---|---|---|---|---|---|
| | | 2020 | 2021 | 2020 | 2021 | 2020 | 2021 |
| Spe1 | *Cyclotella meneghiniana* | 0.18 | 0.16 | 0.09 | 0.11 | 0.09 | 0.1 |
| Spe2 | *Melosira granulata* | 0.13 | 0.11 | 0.04 | 0.03 | 0.05 | 0.08 |
| Spe3 | *Navicula cryptocephala* | 0.06 | 0.04 | | | 0.05 | |
| Spe4 | *Gomphonema acuminatum* | | | 0.04 | 0.09 | | 0.08 |
| Spe5 | *Gomphonema acuminatum* | | | | 0.05 | | |
| Spe6 | *Euglena acus* | | 0.08 | | | | 0.09 |
| Spe7 | *Synedra acus* | 0.07 | | | | 0.06 | |
| Spe8 | *Navicula radiosa* | | | | | 0.03 | |
| Spe9 | *Cosmarium granatum* | | | | | 0.06 | 0.04 |
| Spe10 | *Ceratium hiundinella* | | | | | 0.03 | |

Euglenophyta (39 species), Cyanophyta (17 species), Dinophyta (2 species), Xanthophyta (1 species), and Cryptophyta (2 species). Species number was higher in autumn (79 in 2020, 68 in 2021), moderate in spring (73 in 2020, 54 in 2021), and lower in summer (40 in 2020, 36 in 2021). Ten species were dominant over both years, and two species were dominant in all seasons during the investigation (Table 3).

The seasonal variation of phytoplankton abundance at each sampling sites is shown in Fig 4 and S1 Fig. The phytoplankton abundance varied between seasons and was higher in summer ($613.71 \times 10^4$ ind. L$^{-1}$ in 2020; $591.1 \times 10^4$ ind. L$^{-1}$ in 2021) but lower in autumn ($356.2 \times 10^4$ ind. L$^{-1}$ in 2020; $332.7 \times 10^4$ ind. L$^{-1}$ in 2021) and spring ($186.5 \times 10^4$ ind. L$^{-1}$ in 2020; $147.3 \times 10^4$ ind. L$^{-1}$ in 2021). Meanwhile, the phytoplankton abundance also showed obvious spatiotemporal variations. The sampling site HLS1 had higher phytoplankton abundance than the other sites in summer 2020 ($847.35 \times 10^4$ ind. L$^{-1}$). The highest mean value of phytoplankton abundance ($823.4 \times 10^4$ ind. L$^{-1}$) was recorded at site HLS7, which was near the outfall of a village.

Species such as *Melosira granulata* (Ehrenberg) Ralfs, *Cyclotella meneghiniana* Kutzing, *Navicula cryptocephala* Lange-Bert, and *Pandorina morum* were more abundant in all sites during the study period (S2 Table). The dominant species in HLS1 was *Cyclotella meneghiniana* in 2020 and in spring 2021, but *Navicula cryptocephala* was more abundant in autumn 2021. In HLS2, the dominant species changed from *Melosira granulata* in spring to *Navicula cryptocephala* in summer and autumn 2020; the species *Cyclotella meneghiniana* and *Euglena acus* Ehrenberg were more abundant in 2021. HLS3 was dominated by some diatoms such as *Synedra acus*, *Melosira granulata*, *Gomphonema acuminatum*, and *Navicula radiosa*, *Melosira granulata*, *Cyclotella meneghiniana*, and *Cosmarium granatum* were the dominant species at HLS4.

## Distribution of phytoplankton diversity and evenness

Species diversity was presented by the Shannon-Weaver index ($H'$) is usually used to reflect the complexity of community structure. The more complex the community, the stronger the feedback function to the environment, so that the community structure will be more buffered and tend to be stable [35, 36]. When the phytoplankton diversity index is high, it indicates that environmental conditions are more suitable for the development of phytoplankton species [20, 37, 38].

The Pielou evenness index ($J$) was used to evaluate the evenness of phytoplankton species, and our results showed a positive correlation with species diversity. The Shannon-Weaver index for phytoplankton diversity varied from 1.43 to 3.59 at the sampling sites for 2020–2021

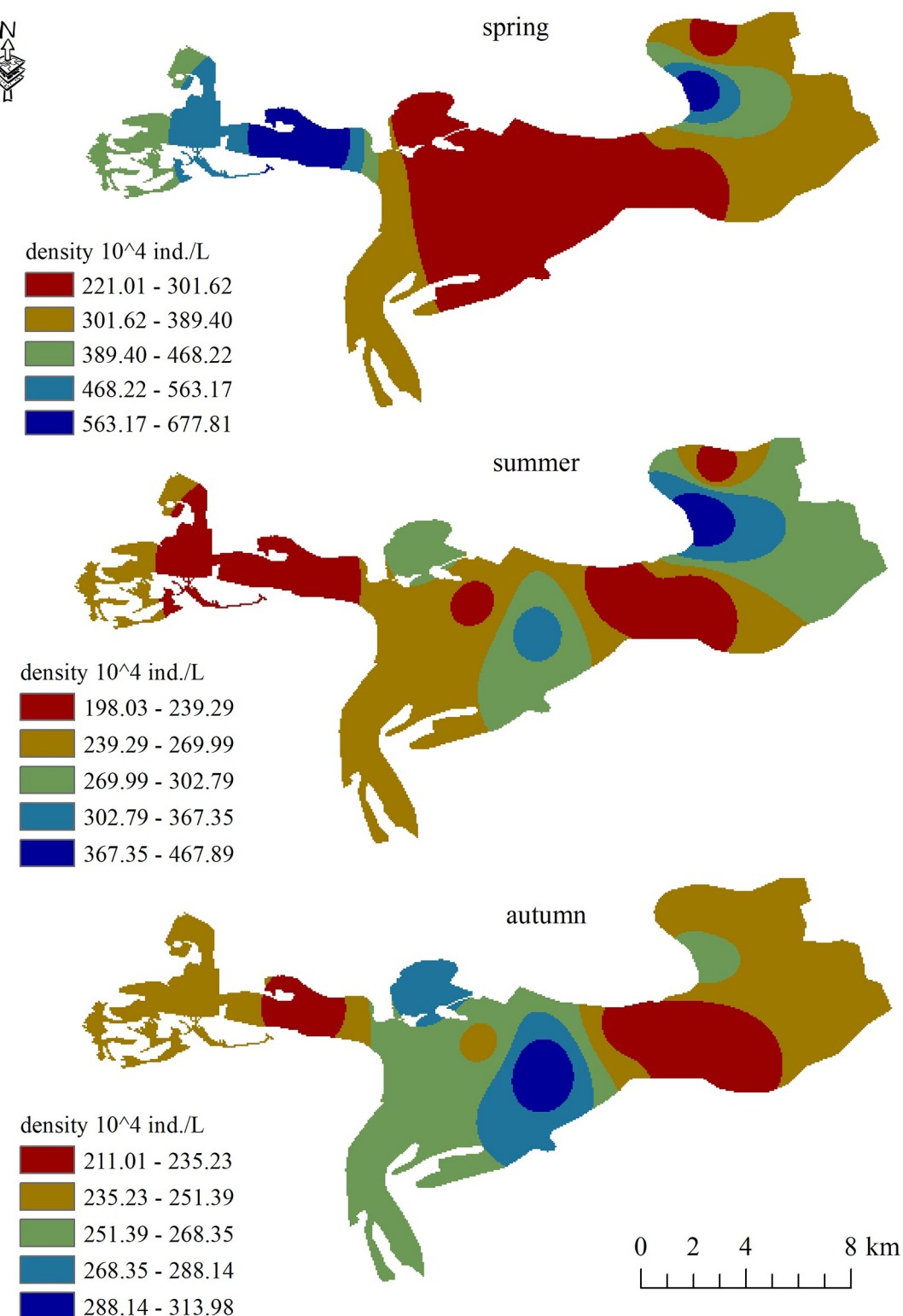

**Fig 4. The distribution of phytoplankton abundance in Hulanhe Wetland.**

**Table 4. Diversity and evenness at thesampling sites in Hulanhe Wetland.**

| | Spring | | | | Summer | | | | Autumn | | | |
|---|---|---|---|---|---|---|---|---|---|---|---|---|
| | 2020 | | 2021 | | 2020 | | 2021 | | 2020 | | 2021 | |
| | H' | J | H' | J | H' | J | H' | J | H' | J | H' | J |
| HLS1 | 1.59 | 0.58 | 1.61 | 0.53 | 2.07 | 0.57 | 2.11 | 0.55 | 1.95 | 0.54 | 1.96 | 0.54 |
| HLS2 | 2.54 | 0.51 | 2.75 | 0.55 | 2.96 | 0.54 | 2.95 | 0.56 | 2.83 | 0.61 | 2.88 | 0.52 |
| HLS3 | 2.56 | 0.56 | 2.80 | 0.57 | 2.96 | 0.57 | 2.93 | 0.54 | 2.94 | 0.63 | 2.94 | 0.61 |
| HLS4 | 2.64 | 0.62 | 2.94 | 0.63 | 3.01 | 0.54 | 3.00 | 0.54 | 2.73 | 0.54 | 2.78 | 0.55 |
| HLS5 | 1.51 | 0.54 | 1.76 | 0.55 | 1.91 | 0.53 | 1.94 | 0.53 | 1.58 | 0.53 | 1.60 | 0.54 |
| HLS6 | 2.50 | 0.57 | 2.61 | 0.59 | 2.90 | 0.61 | 2.89 | 0.66 | 2.73 | 0.65 | 2.74 | 0.65 |
| HLS7 | 3.53 | 0.55 | 3.18 | 0.58 | 3.32 | 0.62 | 3.35 | 0.61 | 3.29 | 0.63 | 3.29 | 0.65 |
| HLS8 | 1.49 | 0.54 | 1.54 | 0.53 | 1.58 | 0.52 | 1.59 | 0.54 | 1.50 | 0.54 | 1.50 | 0.53 |

(Table 4). The highest value for the diversity index appeared at station HLS5 (3.59) in summer 2020, and the lowest was at station HLS8 (1.43) in 2020 and station HLS1 (1.49) in autumn 2021. The Shannon-Weaver index at each sampling site in 2021 was generally higher than that in 2020. The species evenness index varied from 0.51 to 0.68. Both indices were relatively consistent. Average values were lower in 2021 than in 2020. In the current study, the index of diversity (*H'*), and evenness (*J*) in all sampling sites were oligosaprobic and mesosaprobic during the two years in Hulanhe Wetland.

## Relationships between phytoplankton diversity and environmental factors

Redundancy analysis (RDA) was performed to reveal the relationships between the environmental variables and phytoplankton biomass in different sampling sites during two years. Fig 5 shows that the majority of phytoplankton are distributed in small blocks around the intersection of axis 1 and axis 2, showing a decreasing trend from the center to the edge, and in areas that have high correlation with environmental variables, individual species are distributed more intensively. A value of 51.8% of the total cumulative percentage variance of species data was explained and the first two axes were 27.8%. The eigenvalues of axis 1 were higher than that of axis 2, while species-environment correlations in axis 2 (0.998) were the highest among the four axes (Table 5), the variance in the species-environment relationships shows that the axis 1 and axis 2 interpret 38.3%, and 60.6%, respectively. Gradients in environmental variables, which were significantly related to changes in species assemblage composition included concentrations of the nine factors.

Most environmental factors such as WT, TN, TP, pH, BOD, and COD were correlated negatively with the first axis; eigenvalues were –0.0265, –0.0466, –0.0975, –0.2876, –0.3026, and –0.3509, respectively. Axis 2 was correlated positively with most environmental factors except pH and DO. The species were separated into four groups by CCA ordination.

## Discussion

In the current study, two phytoplankton diversity indices (Pielou and Shan-non-Weaver) were used to assess the water quality in Hulanhe Wetland, NE China. Both the phytoplankton diversity indices and *TSI* (Σ) showed high spatial heterogeneity. According to the water quality index, N and P content of the water bodies are high and in the range of mesotrophic level to light eutrophic level. In an earlier study, a contribution to elevated *TSI* (Σ) in Longfeng Wetland, NE China, was attributed to the wastewater discharged by nearby industries and residents [6]. We found that some organic pollutants from the agricultural production process

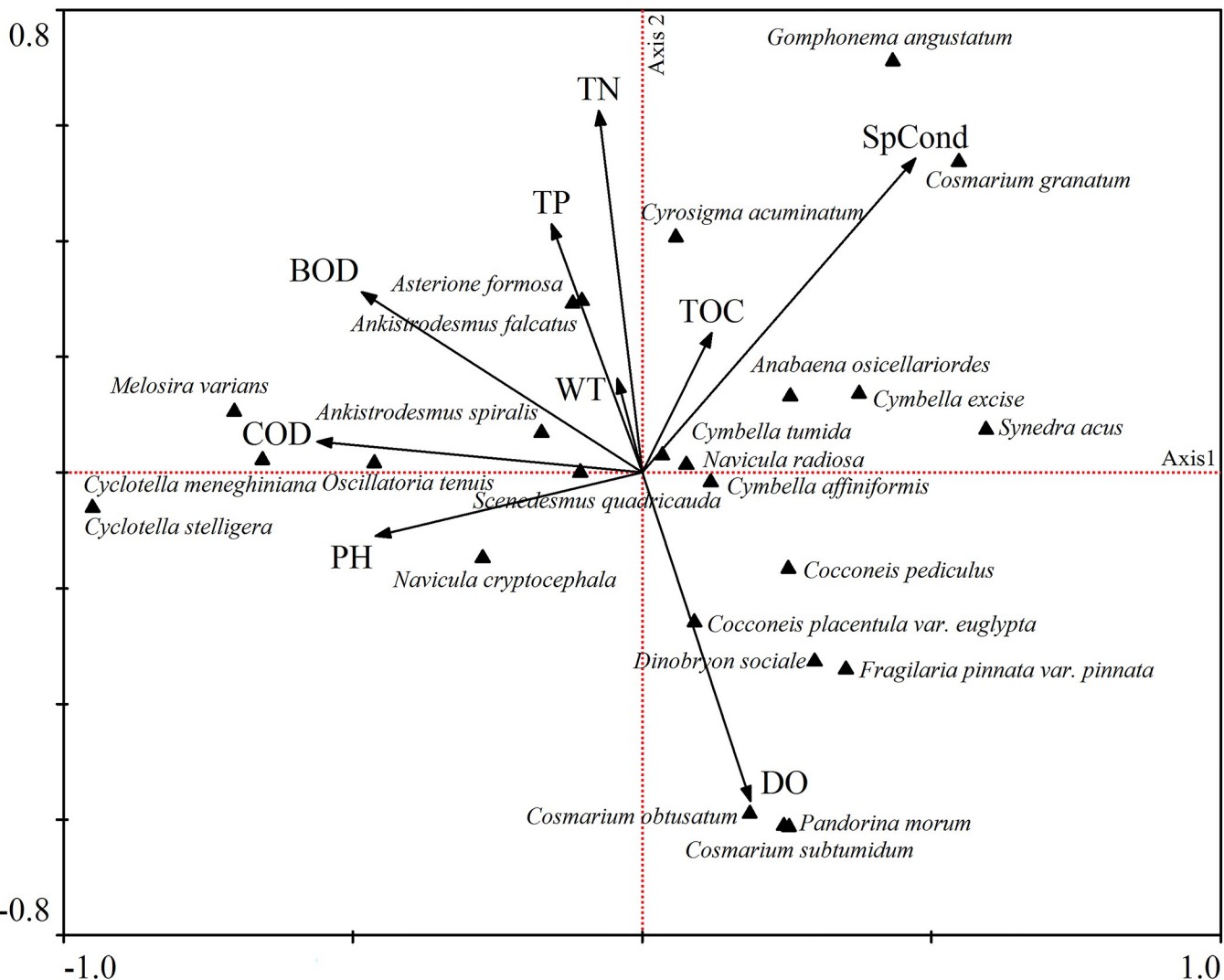

**Fig 5. Redundancy analysis (RDA) of phytoplankton assemblages in Hulanhe Wetland.** *An-ac*: *Ankistrodesmus acicularis*; *An-sp*: *Ankistrodesmus spiralis*; *As-fo*: *Asterionella formaosa*; *Ce-hi*: *Ceratium hirundinella*; *Cl-ac*: *Closterium acerosum*; *Co-gr*: *Cosmarium granatum*; *Cy-ac*: *Cyrosigma acu-minatum*; *Cy-cu*: *Cymbella cuspidate*; *Cy-me*: *Cyclostella meneghiniana*; *Cy-ex*: *Cymbella excise*; *Cy-tu*: *Cymbella tumida*; *Eu-ac*: *Euglena acus*; *Fr-va*: *Fragilaria vaucheriae*; *Go-ac*: *Gomphonema acuminatum*; *Go-an*: *Gomphonema angustatum*; *Go-pa*: *Gomphonema parvulum*; *Me-gr*: *Melosira granulate*; *Me-te*: *Merismopedia tenuissima*; *Na-ca*: *Navicula capitatoradiata*; *Na-vi*: *Navicala viridula*; *Ni-ca*: *Nitzschia cap-itellata*; *Os-te*: *Oscillatoria tenuis*; *Pa-mo*: *Pandorina morum*; *Sc-qu*: *Scenedesmus quadricauda*; *Sy-ac*: *Synedra acus*.

were discharged into the water, so that the COD concentration increased and a large number of pollution-resistant species such as *Scenedesmus quadricauda* and *Asterionella formosa* were found, indicating the organic pollutants related to agricultural can change the phytoplankton

**Table 5. RDA between phytoplankton species and environmental variables for the studied period.**

|  | Axis 1 | Axis 2 | Axis 3 | Axis 4 |
|---|---|---|---|---|
| Eigenvalues | 0.199 | 0.116 | 0.063 | 0.046 |
| Species-environment correlations | 0.973 | 0.998 | 0.963 | 0.961 |
| Cumulative percentage variance of species data | 27.81 | 52.40 | 64.91 | 16.76 |
| Cumulative percentage variance of species-environment relation | 38.32 | 60.64 | 72.72 | 81.63 |

community structure [19, 39, 40]. The present study indicated that additional pollutant sources need to be considered.

Biological characteristics, structure, and function of phytoplankton are obviously different from other plants in wetland aquatic ecosystems [41–43]. In our study, the community structure of phytoplankton showed significant spatial differences, mainly related to the presence of diatoms and green phytoplankton, followed by Euglena and Cyanophyta. Species such as *Melosira granulata*, *Cyclotella meneghiniana*, *Navicula cryptocephala*, and *Pandorina morum* were more abundant in all sites during the study period. Previous studies have found that the extension of water exchange cycles caused by human activities, the accompanying warming effect of climate change, and the decrease in wind speed can all increase the sensitivity of phytoplankton to nutrients [22, 44]. For example, the environmental monitoring area (EMA), due to it being low-lying and a slow water flow (water flow velocity ranged from 0.05 to 0.08 m s$^{-1}$), there is an accumulation of nutrients. More residential area garbage discharged into the water at this location, coupled with sustained wind direction, the prevailing wind direction is SSW and SW in study periods, resulting in some nutritional indicator species blooms [45]. The presence of species *Euglena oxyuris* in FRA (sites HLS7 and HLS8), which had the highest biomass in summer, can be explained by the fact that site HLS8 is located near farmland, they were severely affected by human activities.

Species diversity is a unique biological characteristic of ecological communities: there are community-specific species composition and abundance characteristics. Nutrients can affect the phytoplankton number, conversely, as aquatic ecosystems producers, phytoplankton can also absorb nutrients from the water [46–49]. In Patoucheas's study, phytoplankton are sensitive to various environmental variables, such as nutrient content and temperature, which explain the most variance of phytoplankton, especially the occurrence of diatoms [50]. Furthermore, temperature is the limiting factor of phytoplankton increase [11, 51, 52]. In Rodrigues's research, a sudden increase of phytoplankton in response to both increased winter and summer temperatures represents a major change in the aquatic biota diversity [40]. As the indices revealed in our study, the values for each sampling site were relatively consistent. In summer, the indices were higher indicating that the temperature was higher, the photosynthetic efficiency of phytoplankton was higher, and the wetland water environment was more stable. When the temperature was lower, the species and number of phytoplankton were also reduced, and the environment was relatively unstable. In sampling sites HLS7 and HLS8, some pollution phytoplankton were found such as *Cyclostephanos dubius*, *Euglena gracilis*, and *Microcystis aeruginosa*, and the diversity index and evenness index declined. This is not surprising because this site is affected by human activity, the increase of pollutants generated in life and production leads to deterioration of water bodies, reduction of submerged plants, and a reduction in the ability of the water to self-purify. This type of water environment is only suitable for the survival of pollution-resistant species, with less species composition and diversity, reflecting a poor grade of water quality [24, 38].

Indeed, in the present study, significant correlations were observed between the phytoplankton diversity indices and *TSI* (Σ) in the Hulanhe Wetland (Fig 6). This conflicts with the previous finding that for water quality assessment, phytoplankton diversity indices were superior to water quality monitoring based on physicochemical analyses [37, 53, 54]. In contrast, our result is consistent with the conclusions of other studies conducted in different water bodies. For example, Zhang evidenced that in Zhalong Wetland, variation in diatom diversity indices was always associated with water quality changes [55]. Furthermore, Huo studied phytoplankton community assembly in a mid-sized river over different hydrological periods and found that both spatial and environmental variables had high explanatory power on the community structure [56]. Therefore, phytoplankton diversity can effectively reflect water

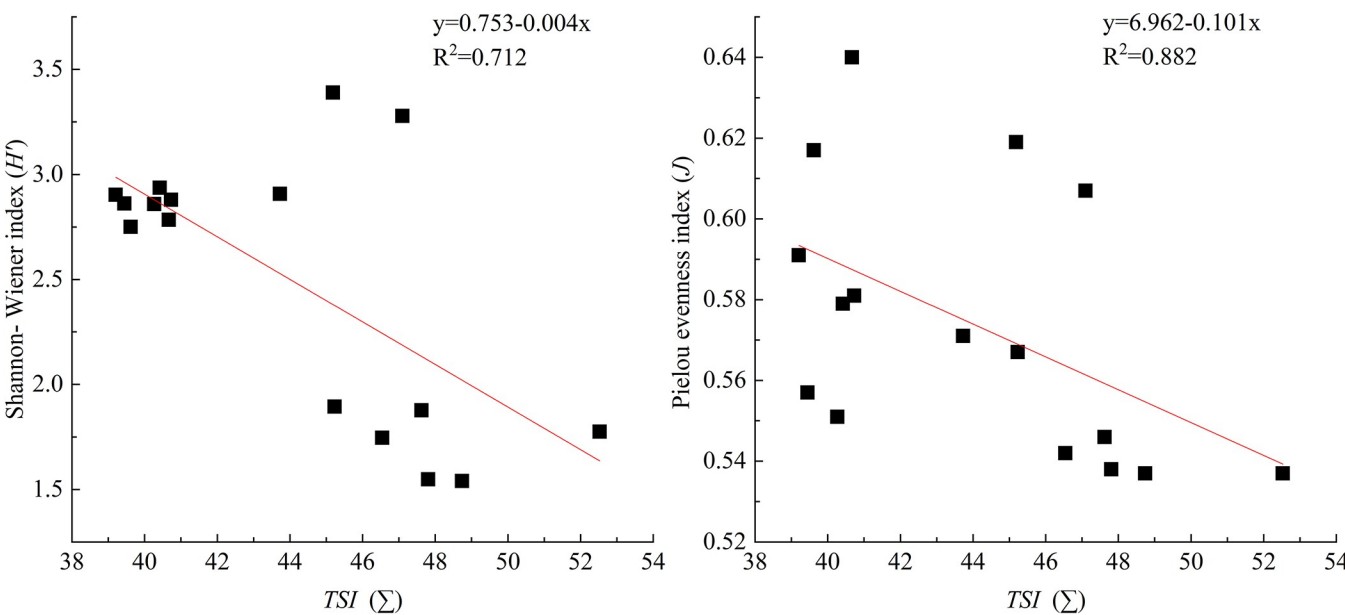

**Fig 6. Relationships between TSI and Shannon-Weaver, Pielou indices of phytoplankton in Hulanhe Wetland.**

quality in freshwater wetland. However, lower correlations between phytoplankton diversity and water quality parameters have also been found in the Yellow River and the mid-lower Yangtze Lakes [21, 57], indicating that significant differences in ecological background and complex terrain are the main reasons for the mismatched.

The results of RDA and CCA showed that differences reflected variations in the conditions of the environmental variables. Some environmental variables such as TN, TP, and DO, all of which correlated with assemblage changes, were markedly different among the sampling sites (S2 Fig). Significant correlation was represented by the distribution of phytoplankton and nutrients in the Hulanhe Wetland. The *TSI* ($\sum$) indicated that the water was in mesotrophic state. *Scenedesmus quadricauda*, *Gomphonema acuminatum*, *Cyclotella meneghiniana*, and *Oscillatoria tenuis Agardh ex Gomont* are good indicators of TP, which mainly occurred in EMA and FRA. Oscillatoria tenuis was reported as a potential indicator for phytoplankton blooms in other wetlands, lakes, and rivers in Northeast China, such as Zhalong Wetland, Genhe River in the Greater Hinggan Mountains, while *Cyclotella meneghiniana* is a eutrophic species, with a wide tolerance to temperature fluctuations and corresponds to higher TP values [50, 58]. *Melosira granulata* and *Euglena oxyuris* are good indicators of TN, mainly distributed in the sites with high concentration of TN. The growth of phytoplankton is closely related to the content of nutrients in water [59–61].

Due to the changes in nutrients, the number of trophic levels in the food chain will change. The phytoplankton community structure affects the physical and chemical factors of the water body, such as transparency, suspended solids, and the pH value [62–64]. Studying relationship between phytoplankton and environmental factors can be an advantage in studying different areas with phytoplankton diversity. Phytoplankton diversity increases from mesotrophic to meso-eutrophic lakes and wetlands, which was also observed in our study [65–69]. We also found a significant number of eutrophic taxa, which shows that most areas in the Hulanhe Wetland are heavily impacted by human activities, the main activity being the use of organic fertilizer during the agricultural production process.

## Conclusions

In this study, *TSI* (Σ) values and phytoplankton diversity were used to assess the water quality. Our results revealed that the water quality was oligosaprobic and mesosaprobic in Hulanhe Wetland. Especially, the relationships between each phytoplankton diversity index and *TSI* (Σ) fit well with the standard curves of water quality classification that were generated based on the respective biodiversity index and *TSI* (Σ) values. In addition, nutrient content (TN and TP) and temperature were important environmental variables affecting the phytoplankton community structure. Therefore, it is suggested that phytoplankton diversity is a suitable water quality indicator, it can reflect habitat changes to a certain extent.

## Supporting information

**S1 Table.** *TSI(BOD), TSI(COD), TSI(TN), TSI(TP)* **in different sampling sites.**
(PDF)

**S2 Table. Phytoplankton assemblages used for redundancy analysis (RDA).**
(PDF)

**S1 Fig.** Assemblage composition and relative abundance of the most common species in eight sites (a. abundance of the species in 2020; b. abundance of the species in 2021).
(TIF)

**S2 Fig.** Detrended correspondence analysis (DCA) of algae assemblages in different sites during two years (a. DCA in 2020; b. DCA in 2021).
(TIF)

**S1 Data. Ordinary Kriging of phytoplankton abundance spatial distribution.**
(MPK)

## Acknowledgments

The authors thank Dr. Yawen Fan, Xinxin Lu, Fengyang Sui (College of Life Science and Technology, Harbin Normal University) for their insightful comments and helpful suggestions.

## Author Contributions

**Conceptualization:** Hongkuan Hui, Yinxin Wei.

**Formal analysis:** Dan Su, Haitao Zhou, Zirui Peng.

**Funding acquisition:** Hongkuan Hui, Xiao Liu.

**Investigation:** Hongkuan Hui, Dan Su.

**Writing – original draft:** Hongkuan Hui, Haitao Zhou, Zirui Peng.

**Writing – review & editing:** Yinxin Wei, Dan Su.

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
