## [Decision Letter · Decision Letter 0]

29 Apr 2024

PONE-D-24-12760Ecological assessment of water quality in freshwater wetlands based on the effect of environmental heterogeneity on phytoplankton communities in Northeast ChinaPLOS ONE

Dear Dr. Hui,

Thank you for submitting your manuscript to PLOS ONE. After careful consideration, we feel that it has merit but does not fully meet PLOS ONE’s publication criteria as it currently stands. Therefore, we invite you to submit a revised version of the manuscript that addresses the points raised during the review process.

We look forward to receiving your revised manuscript.

Kind regards,

Dharmendra Kumar Meena

Academic Editor

PLOS ONE

Journal Requirements:

 When submitting your revision, we need you to address these additional requirements. 1. Please ensure that your manuscript meets PLOS ONE's style requirements, including those for file naming. The PLOS ONE style templates can be found at https://journals.plos.org/plosone/s/file?id=wjVg/PLOSOne_formatting_sample_main_body.pdf and https://journals.plos.org/plosone/s/file?id=ba62/PLOSOne_formatting_sample_title_authors_affiliations.pdf 2. In your Methods section, please provide additional information regarding the permits you obtained for the work. Please ensure you have included the full name of the authority that approved the field site access and, if no permits were required, a brief statement explaining why. 3. Thank you for stating in your Funding Statement: "Funding acquisition: Hongkuan Hui, Xiao Liu.This work was supported by the Natural Science Foundation of Shandong Province (ZR2023QC238, ZR2023QC253), the Project of Shandong Province Higher Educational Science and Technology Program (No. J18KA199)." Please provide an amended statement that declares *all* the funding or sources of support (whether external or internal to your organization) received during this study, as detailed online in our guide for authors at http://journals.plos.org/plosone/s/submit-now.  Please also include the statement “There was no additional external funding received for this study.” in your updated Funding Statement. Please include your amended Funding Statement within your cover letter. We will change the online submission form on your behalf. 4. PLOS requires an ORCID iD for the corresponding author in Editorial Manager on papers submitted after December 6th, 2016. Please ensure that you have an ORCID iD and that it is validated in Editorial Manager. To do this, go to ‘Update my Information’ (in the upper left-hand corner of the main menu), and click on the Fetch/Validate link next to the ORCID field. This will take you to the ORCID site and allow you to create a new iD or authenticate a pre-existing iD in Editorial Manager. Please see the following video for instructions on linking an ORCID iD to your Editorial Manager account: https://www.youtube.com/watch?v=_xcclfuvtxQ 5. Please amend the manuscript submission data (via Edit Submission) to include author Zirui Peng. 6. We note that Figures 1 and 4 in your submission contain map/satellite images which may be copyrighted. All PLOS content is published under the Creative Commons Attribution License (CC BY 4.0), which means that the manuscript, images, and Supporting Information files will be freely available online, and any third party is permitted to access, download, copy, distribute, and use these materials in any way, even commercially, with proper attribution. For these reasons, we cannot publish previously copyrighted maps or satellite images created using proprietary data, such as Google software (Google Maps, Street View, and Earth). For more information, see our copyright guidelines: http://journals.plos.org/plosone/s/licenses-and-copyright. We require you to either (1) present written permission from the copyright holder to publish these figures specifically under the CC BY 4.0 license, or (2) remove the figures from your submission: a. You may seek permission from the original copyright holder of Figures 1 and 4 to publish the content specifically under the CC BY 4.0 license.   We recommend that you contact the original copyright holder with the Content Permission Form (http://journals.plos.org/plosone/s/file?id=7c09/content-permission-form.pdf) and the following text:“I request permission for the open-access journal PLOS ONE to publish XXX under the Creative Commons Attribution License (CCAL) CC BY 4.0 (http://creativecommons.org/licenses/by/4.0/). Please be aware that this license allows unrestricted use and distribution, even commercially, by third parties. Please reply and provide explicit written permission to publish XXX under a CC BY license and complete the attached form.” Please upload the completed Content Permission Form or other proof of granted permissions as an ""Other"" file with your submission. In the figure caption of the copyrighted figure, please include the following text: “Reprinted from [ref] under a CC BY license, with permission from [name of publisher], original copyright [original copyright year].” b. If you are unable to obtain permission from the original copyright holder to publish these figures under the CC BY 4.0 license or if the copyright holder’s requirements are incompatible with the CC BY 4.0 license, please either i) remove the figure or ii) supply a replacement figure that complies with the CC BY 4.0 license. Please check copyright information on all replacement figures and update the figure caption with source information. If applicable, please specify in the figure caption text when a figure is similar but not identical to the original image and is therefore for illustrative purposes only.The following resources for replacing copyrighted map figures may be helpful: USGS National Map Viewer (public domain): http://viewer.nationalmap.gov/viewer/The Gateway to Astronaut Photography of Earth (public domain): http://eol.jsc.nasa.gov/sseop/clickmap/Maps at the CIA (public domain): https://www.cia.gov/library/publications/the-world-factbook/index.html and https://www.cia.gov/library/publications/cia-maps-publications/index.htmlNASA Earth Observatory (public domain): http://earthobservatory.nasa.gov/Landsat: http://landsat.visibleearth.nasa.gov/USGS EROS (Earth Resources Observatory and Science (EROS) Center) (public domain): http://eros.usgs.gov/#Natural Earth (public domain): http://www.naturalearthdata.com/ 7. We are unable to open your Supporting Information file 
Supporting information.rar. Please kindly revise as necessary and re-upload.

Additional Editor Comments:

article is recommended for minor revision

Reviewers' comments:

Reviewer's Responses to Questions

**Comments to the Author**

1. Is the manuscript technically sound, and do the data support the conclusions?

Reviewer #1: Yes

Reviewer #2: Yes

2. Has the statistical analysis been performed appropriately and rigorously? 

Reviewer #1: Yes

Reviewer #2: Yes

3. Have the authors made all data underlying the findings in their manuscript fully available?

Reviewer #1: Yes

Reviewer #2: No

4. Is the manuscript presented in an intelligible fashion and written in standard English?

Reviewer #1: Yes

Reviewer #2: Yes

5. Review Comments to the Author

Reviewer #1: Dear author, it was nice work but i suggest to develop it using IBI (Intgrated Bioloigcal Indecies) i was use it since 2010

i think mixed your work with IBI will be wounderfull

good job and regards

Reviewer #2: Based on the analysis and review of the manuscript, here are the review comments and recommendations provided to the author:

Technical Soundness:

The study appears to be technically sound, utilizing standard protocols and procedures for sampling, analysis, and diversity calculations. The methods section offers a detailed description of the sampling and analysis techniques, indicating a robust and replicable methodology.

Statistical Analysis:

The statistical analysis approaches employed in the study seem appropriate for examining the relationships between environmental variables and phytoplankton community structure. The authors followed standard protocols, suggesting that the analysis is statistically robust.

Data Availability:

The manuscript does not explicitly mention data deposition in repositories. It is recommended that the authors deposit the underlying data in an appropriate public repository, adhering to relevant guidelines, to ensure the data are accessible and reusable by the research community.

Manuscript Presentation:

The manuscript is well-structured and written, with a clear introduction, methods, results, discussion, and conclusion. However, the authors could enhance the accessibility of the manuscript by:

1. Addressing data deposition, as mentioned above.

2. Clarifying the availability of any software or code used during the study.

3. Examining whether the manuscript contains any experiments of Dual Use Concern, as defined by NIH, and addressing data deposition in repositories.

6. PLOS authors have the option to publish the peer review history of their article (what does this mean?). If published, this will include your full peer review and any attached files.

Reviewer #1: **Yes: **Ibrahim M. Al-Sudani (Iraq)

Reviewer #2: **Yes: **Wisdom Adzigbli

---

## [Author Response · Author response to Decision Letter 0]

3 Jun 2024

Responses to Reviewers

Manuscript Number: PONE-D-24-12760

Title: Ecological assessment of water quality in freshwater wetlands based on the effect of environmental heterogeneity on phytoplankton communities in Northeast China 

Journal: PLOS ONE

Dear Dharmendra Kumar Meena, Academic Editor of PLOS ONE

We are truly grateful to you for giving us the chance to revise the manuscript. At the same time, we would like to thank you and the reviewers for your critical comments and thoughtful suggestions. Based on these comments and suggestions, we have made careful modifications to the original manuscript. All changes made to the text are in red. We hope the new manuscript will meet your journal’s standard.

Thanks to the reviewers for the thoughtful and thorough review. We have considered all the specific comments carefully and revised the manuscript accordingly. All the questions raised are answered point by point as follows.

Response: Thank you very much for your reminder. We have made revisions and improvements to the manuscript to ensure meets PLOS ONE's style requirements (including file naming, images, etc.)

Response: No permits are required to enter the research site. We have established long-term cooperation with the Hulanhe Wetland Conservation Area.

"Funding acquisition: Hongkuan Hui, Xiao Liu.This work was supported by the Natural Science Foundation of Shandong Province (ZR2023QC238, ZR2023QC253), the Project of Shandong Province Higher Educational Science and Technology Program (No. J18KA199)."

Response: Thank you for bringing this to our attention. This work was supported by the Natural Science Foundation of Shandong Province (ZR2023QC238, ZR2023QC253), the Project of Shandong Province Higher Educational Science and Technology Program (No. J18KA199). There was no additional external funding received for this study.

Response: I have an ORCID iD (https://orcid.org/0009-0003-3066-2988) and that it is validated in Editorial Manager.

5. Please amend the manuscript submission data (via Edit Submission) to include author Zirui Peng.

Response: Thank you for bringing this to our attention, we have modified the information in the system and added the author Zirui Peng.

6. We note that Figures 1 and 4 in your submission contain map/satellite images which may be copyrighted. All PLOS content is published under the Creative Commons Attribution License (CC BY 4.0), which means that the manuscript, images, and Supporting Information files will be freely available online, and any third party is permitted to access, download, copy, distribute, and use these materials in any way, even commercially, with proper attribution. For these reasons, we cannot publish previously copyrighted maps or satellite images created using proprietary data, such as Google software (Google Maps, Street View, and Earth). For more information, see our copyright guidelines: http://journals.plos.org/plosone/s/licenses-and-copyright.

a. You may seek permission from the original copyright holder of Figures 1 and 4 to publish the content specifically under the CC BY 4.0 license. 

Response: We are very grateful to your critical comments and thoughtful suggestions. We have contacted the Map Data Supply Center, whose staff told us that map data (Figures 1) can be referenced according to the data reference method provided on the official website (www.resdc.cn) to meet the data release requirements (https://www.resdc.cn/DOI/DOI.aspx?DOIID=121). Therefore, we revised the manuscript according to its requirements. We have visited all the map websites you provided, but we are sorry that the map data provided by these websites cannot be reviewed by the Map Technology Review Center of the Ministry of Natural Resources, so it is not applicable to the drawing of the manuscript map. Figure 4 does not involve the original copyright license and is a figure processed by the author based on open resources.

7. We are unable to open your Supporting Information file Supporting information.rar. Please kindly revise as necessary and re-upload.

Response: We sincerely thank the reviewer for careful reading. We have made modifications to Supporting Information file "Supporting information.rar" format and have resubmitted it. S1 Data, which is a file with the suffix ". emk", can be opened using ArcGIS software.

Response: Thank you for bringing this to our attention. After careful review of the references, we have not cited papers that have been retracted.

Reviewer #1:

1. Dear author, it was nice work but i suggest to develop it using IBI (Intgrated Bioloigcal Indecies) i was use it since 2010, i think mixed your work with IBI will be wounderfull good job and regards.

Response: Thanks for your valuable suggestions. All of your suggestions are very important. We are interested in using IBI (Integrated Bioacoustic Industries) to evaluate water quality and have consulted some literature, such as “Mohamed, A. R. M., Hussain, N. A., Al Noor, S. S., Mutlak, F. M., Al Sudani, I. M.,&Mojer, A. M. (2012). Ecological and biological aspects of fish assemblage in the Chybayish marsh, Southern Iraq[J]. Ecohydrology & Hydrobiology, 12 (1), 65-74." However, we made a full consideration for this manuscript, we suggested the current work was sufficient to solve our scientific problems. Of course, we will consider combining IBI for next research.

Reviewer #2:

1. Based on the analysis and review of the manuscript, here are the review comments and recommendations provided to the author:

Technical Soundness:

The study appears to be technically sound, utilizing standard protocols and procedures for sampling, analysis, and diversity calculations. The methods section offers a detailed description of the sampling and analysis techniques, indicating a robust and replicable methodology.

Response: Thank you again for your positive comments and valuable suggestions to improve the quality of our manuscript.

Statistical Analysis:

The statistical analysis approaches employed in the study seem appropriate for examining the relationships between environmental variables and phytoplankton community structure. The authors followed standard protocols, suggesting that the analysis is statistically robust.

Response: We feel great thanks for your professional review work on our manuscript.

Data Availability:

The manuscript does not explicitly mention data deposition in repositories. It is recommended that the authors deposit the underlying data in an appropriate public repository, adhering to relevant guidelines, to ensure the data are accessible and reusable by the research community.

Response: We think this is an excellent suggestion. The datasets used or analyzed during the current study are available from the corresponding author on reasonable request. Supporting information is also provided in the PLOS ONE submission system. 

Manuscript Presentation:

The manuscript is well-structured and written, with a clear introduction, methods, results, discussion, and conclusion. However, the authors could enhance the accessibility of the manuscript by:

1. Addressing data deposition, as mentioned above.

Response: We sincerely thank the reviewer for careful reading. We have resubmitted the relevant data to improve the accessibility of the manuscript.

2. Clarifying the availability of any software or code used during the study.

Response: We feel great thanks for your professional review work on our article. Most of the software used in our article is open source and trial versions.

3. Examining whether the manuscript contains any experiments of Dual Use Concern, as defined by NIH, and addressing data deposition in repositories.

Response: Thanks for your suggestion. Our article does not involve any experiments of Dual Use Concern, as defined by NIH, and we resubmitted the data to improve its accessibility.

Ecological Assessment of Water Quality in Freshwater Wetlands: Effect of Environmental Heterogeneity on Phytoplankton Communities in Northeast China 

Abstract: 

The study delves into how phytoplankton diversity can act as a mirror for water quality within the Hulanhe Wetland. It pinpoints nutrient levels (TN and TP) and temperature as pivotal environmental factors shaping the phytoplankton community's structure. This work holds significance for aquatic ecology and environmental assessment, offering insights crucial for the efficient monitoring and management of wetland ecosystems. 

Response: Thanks very much for your comment, which is highly appreciated.

Introduction: 

The introduction lays the groundwork by highlighting why understanding phytoplankton diversity matters in assessing water quality in freshwater wetlands. It also discusses existing research on using phytoplankton as indicators of water quality and the various factors influencing phytoplankton communities. 

Response: Thanks very much for your comment, which is highly appreciated.

Methods: 

The methods section meticulously describes the sampling and analysis procedures, covering everything from water quality measurements to phytoplankton identification, enumeration, and diversity index calculations. The authors followed standard protocols, ensuring a robust and replicable methodology. 

Response: Thanks very much for your comment, which is highly appreciated.

Results: 

The results section presents the study's findings, particularly focusing on how nutrient levels and temperature affect phytoplankton communities in the Hulanhe Wetland. It emphasizes the link between environmental variables and phytoplankton diversity. 

Response: Thanks very much for your comment, which is highly appreciated.

Discussion: 

In the discussion, the results are put into context within the existing body of knowledge. The implications of the findings for understanding water quality in freshwater wetlands are explored, with an eye towards improving wetland ecosystem monitoring and management strategies. 

Response: Thanks very much for your comment, which is highly appreciated.

Conclusion: 

The conclusion summarizes the main findings regarding how environmental factors influence phytoplankton diversity. It underscores the study's importance for advancing knowledge in aquatic ecology and environmental assessment. 

Response: Thanks very much for your comment, which is highly appreciated.

Recommendation: 

I would encourage the authors to submit a revised version of their manuscript, as their work shows promise in contributing to the field. Areas for improvement could include addressing data deposition in repositories, adhering to relevant guidelines, making the manuscript more accessible to non-specialists, and clarifying the availability of any software created during the study. Additionally, assessing whether the manuscript contains any experiments of Dual Use concern, as defined by NIH, would be beneficial.

Response: Thanks very much for your comment, which is highly appreciated. We have carefully scrutinized the manuscript and made corresponding revisions including Dual Use concern, software created, etc.

---

## [Editor Report · Decision Letter 1]

17 Jun 2024

Ecological assessment of water quality in freshwater wetlands based on the effect of environmental heterogeneity on phytoplankton communities in Northeast China

PONE-D-24-12760R1

Dear Dr. Hui

We’re pleased to inform you that your manuscript has been judged scientifically suitable for publication and will be formally accepted for publication once it meets all outstanding technical requirements.

Kind regards,

Dharmendra Kumar Meena

Academic Editor

PLOS ONE

Additional Editor Comments (optional):

the article can be accepted for publication
---

## [Editor Report · Acceptance letter]

26 Jun 2024

PONE-D-24-12760R1 

PLOS ONE

Dear Dr. Hui, 

I'm pleased to inform you that your manuscript has been deemed suitable for publication in PLOS ONE. Congratulations! Your manuscript is now being handed over to our production team.

Kind regards, 

on behalf of

Dr. Dharmendra Kumar Meena 

Academic Editor

PLOS ONE